# Using Sentiment Analysis in Understanding the Information and Political Pluralism under the Chilean New Constitution Discussion

Cristóbal Balbontín [1,†], Sergio Contreras [2,*,†] and Rodrigo Browne [3,†]

1 Facultad de Ciencias Jurídicas y Sociales, Universidad Austral de Chile, Valdivia 5110566, Chile
2 Escuela de Ciencias Empresariales, Universidad Catolica del Norte, Coquimbo 1780000, Chile
3 Instituto de Comunicación Social, Facultad de Filosofía y Humanidades, Universidad Austral de Chile, Valdivia 5110566, Chile
* Correspondence: scontreras@ucn.cl
† All authors contributed equally to this work.

**Abstract:** There is evidence of constitutional rank in Chile, not only a sectoral rank, to estimate that the regulation of digital media is in an initial phase and thus insufficient to properly protect information pluralism such as political pluralism. This study aims to investigate forms of concentration, such as the communicational flow of digital media, to determine the opportunities and information defects of these media due to regulatory deficiencies in this sector. Data collection was carried out through a qualitative and quantitative methodology. The prospect of the imminent constituent process in Chile provides the opportunity to evaluate possibilities and propose changes not only at the legislative level but also at the constitutional level, which are likely to provide benefits such as freedom of expression, with greater guarantees toward the pluralism of digital media. The latter also means assessing the relevance of enshrining the right to communication in the future.

**Keywords:** digital media; pluralism; freedom of expression; new constitution; web scrapping; sentiment analysis





## 1. Introduction

Political pluralism is in solidarity with information pluralism, essential for fully exercising the right to freedom of expression and the existence of a democratic society. For this purpose, the specialized literature (Zárate-Rojas 2016) distinguishes three types of information pluralism. On the one hand, internal pluralism relates to the information offered by a particular medium, which must ensure that all voices are incorporated when providing information. On the other hand, external pluralism promotes the diversity of sources of information and a plurality of actors related to the media. This category includes state subsidies to promote the creation of new media or strengthen existing ones. Finally, structural pluralism refers to the pluralism of information associated with organizing a diversity of events in the market. It includes the prohibition of the state monopoly and the mechanisms for promoting competition in the information market to avoid concentration in the ownership of news companies.

Although the Chilean legal system, which has internal and structural pluralism elements, has recently been perfected with new legislative initiatives, severe criticism remains regarding the concentration of the media in the country (Ruiz-Tagle Vial 2011). The emergence of digital media undoubtedly offers a relevant opportunity to break media concentration in Chile. However, we hypothesize that these new technologies still have regulatory difficulties that hinder the proper pluralism of information, and, consequently, the political pluralism typical of a democratic society. In this context, this article seeks to reflect on the emergence of a digital media ecosystem in Chile and its constitutional and

legal guarantees to achieve a greater presence of internal, external, and structural pluralism in a space of permanent adaptations to technological changes. In this paper, sentiment analysis is used due to is one of the techniques that can be used to determine pluralism in a certain population such as Cao et al. (2022) and Rao et al. (2022). This technique is used as a proxy for diversity and neutrality and thus avoids the echo chamber effect on media (Cinelli et al. 2021). A number of hashtags were selected based on the trending topics associated with the constitutional discussion along with election candidates' hashtags (e.g., Boric and Kast). The repetition of some hate or negative words could influence the voters through certain outlets (traditional or electronic) who might impose their agenda and therefore would not be neutral to the election process (Cooper et al. 2019; Dwivedi et al. 2021; Kumar et al. 2018; Pandiaraj et al. 2021).

Therefore, our work has the following specific research goals: (i) describe the digital media ecosystem based on external and structural pluralism; (ii) register the digital media ecosystem and its interaction mechanisms with the traditional information system in the formation of Chilean public opinion, based on internal pluralism; (iii) propose elements of discussion for a legislative and constitutional debate that facilitates the generation of a digital ecosystem of a plural nature.

To this end, this project is based on an interdisciplinary and mixed methodology (qualitative and quantitative). In the Section 1, we examine the current state of Chilean legislation to determine the types of information pluralism the country adheres to. In the Section 2, we explore the relationship between information pluralism and the right to communicate as an expression of a democratic society governed by law. In the Section 3, we examine the opportunity offered by digital media to achieve greater information pluralism. For this purpose, we review the methodology applied and the results obtained. The conclusion sheds light on whether the emergence of a digital media ecosystem, together with the existing regulatory guarantees, effectively guarantees greater information pluralism when shaping political public opinion in the country.

## 2. Information Pluralism in Current Chilean Legislation

The framework for Chilean legislation is the Constitution of 1980, which is committed to a model of internal pluralism. Indeed, Article 19 No. 12, while guaranteeing the freedom to express an opinion, enshrines editorial freedom by explicitly stating that freedom is "to inform, without prior censorship, in any form, and in any media". However, it also contains elements of structural pluralism, establishing the prohibition of state monopoly over the media in the second paragraph of Article 19 No. 12 of the Constitution. Although it specifically enshrines a modality of informative pluralism, its version of structural pluralism is solidary of social pluralism, which is recognized in Article 1, third paragraph, of the Constitution: "The State recognizes and protects the intermediate groups through which society is organized, and structured, and guarantees them adequate autonomy to fulfill their own specific purposes"). Consistent with this premise, Article 19 No. 12, in the fourth paragraph of the Constitution, enshrines the right of every person to "found, edit, and maintain newspapers, magazines, and newspapers, under the conditions established by law". The impulse of information pluralism then falls on individuals consistent with the principle of constitutional subsidiarity.

Although in the 1970s, radio and the emerging television were presented as an opportunity to increase structural pluralism, the Constitution originally opted for a more conservative statute against the "cultural industry" that television could be (Bourdieu 2000; Horkheimer et al. 1987) by establishing that only "The State, those universities, and other persons or entities that the law determines, may establish, operate, and maintain television stations".

With the return to democracy, various normative texts advanced in specifically enshrining information pluralism. However, these normative innovations did not abandon the premise of internal pluralism based on editorial freedom. Based on the above, a series of authors argued the need to discipline the concentration of the media, subjecting them to democratic control through state intervention (García 1997). For this purpose, the draft of

the current law No. 19,733 (Press Law) discussed the possibility of establishing maximum quotas in media ownership, thus forcing fragmentation of the owners as a way to attack concentration. This resulted in the intervention of the Constitutional Court, which by ruling No. 226 of 30 October 1995, declared this provision unconstitutional, stating that the phrase contained in the draft that established the state's legal obligation to ensure "the effective expression of the various currents of opinion, as well as the social, cultural, and economic variety of the regions" interfered with the autonomy of those intermediate bodies or associative groups that are the means of social communication, thus violating Article 1, third paragraph, of the Constitution. Thus, despite legislative attempts, Chile maintains a high media concentration. In fact, El Mercurio SAP dominates 53% of the written press in Chile. On the other hand, the Copesa S.A. group controls 46%. Between these two media business giants, they manage 99% of the written press. The remaining 1% belongs to La Nación, Publimetro, and other newspapers (Balbontín and Maldonado 2019, p. 641). Regarding radio, in Chile, there are around 2000 radio concessions, from small local radio stations to radio stations with national coverage, of which approximately 300 belong to national and foreign business groups (Ramírez-Cáceres 2009) Regarding open television, there are six major television channels in Chile: First is Televisión Nacional de Chile, a state-owned channel. Then, there is Canal 13, which, although initially owned by the Universidad Católica de Chile, was fully purchased by the Luksic group in 2017. Another channel is TV+, co-owned by the business group TV+ SpA and PUCV Multimedios SpA. Finally, we have Chilevisión, which occupies channel 11; Canal Mega, co-owned by the Bethia group and the holding company Discovery Communications, which occupies channel 9; and Canal La Red, which occupies channel 4. Regarding the audience, in 2018, the statistical yearbook of supply and consumption of the National Television Council indicated that the Mega channel had the most audience, with 32.4%, followed by Chilevisión with 21.7%, Canal 13 with 21.3%, Televisión Nacional de Chile with 16.7%, La Red with 5.9%, and finally UCV TV (now TV +) with 1.9% (Consejo Nacional de Televisión 2019). Historically, the problem was referred to as the existing antitrust legislation, so the premise of ensuring the pluralism of information is no different from the one that operates to ensure a perfect market for any good or service.

Thirteen years later, while maintaining its defense of editorial freedom, the Constitutional Court was established to endorse the constitutionality of broadcasting information defined by the state and to preserve portions of the radio electric space for communities, and/or regional and local organizations. This decision was made by articulating the right to information with the right of people to participate with equal opportunities in national life (Article 1, paragraph 5 of the Constitution). Along with the above, Law No. 20,433 creates citizen community broadcasting services. It establishes in Article 9—compared to radio stations with maximum coverage—that only non-profit, private legal persons may hold a concession if their essential purposes include promoting the general interest through the pursuit of specific objectives of a civic, social, cultural, or spiritual nature. Such persons must be constituted in and reside in Chile. The same applies to Law No. 20,750 on digital television, which establishes a mechanism for reserving the broadcast frequencies of regional and local ranges while limiting the number of concessions and the duration of television concessions. On the other hand, Article 13 of the General Telecommunications Law limits the accumulation of frequencies by the same company or economic group for the same location, which precisely seeks to avoid media concentration.

## 3. Digital Media: An Opportunity for Information Pluralism?

In line with structural pluralism, digitization undoubtedly offers a significant opportunity to break media concentration in Chile. The Chilean Supreme Court itself, hearing an appeal for protection (case No. 450-2018), has understood that social networks such as Twitter, Facebook, or YouTube are social media under the broad definition of social media in Article 2 of Law No. 19733. In terms of the difference between social and digital media, Article 2 of Law No. 19,733 in Chile defines social media as "those capable of transmitting, divulging, disseminating or

propagating, on a stable and periodic basis, texts, sounds or images intended for the public, regardless of the medium or instrument used." Digital media is therefore a type of social media that is characterized by three processes, which are digitization, convergence, and the use of the Internet (Forte et al. 2012; Ramírez-Cáceres 2021)

However, the existence of these new technological media does not necessarily ensure information pluralism (Balbontín and Maldonado 2019). Many observations and policy recommendations for Chile have been made by international organizations to ensure that new technologies can consolidate information pluralism. The UN Special Rapporteur, in conjunction with the Representative on Freedom of the Media of the Organization for Security and Cooperation in Europe (OSCE); the Special Rapporteur for Freedom of Expression of the Inter-American Commission on Human Rights of the OAS; and the Special Rapporteur on Freedom of Expression and Access to Information of the African Commission on Human and Peoples' Rights (ACHPR) expressed in a joint statement that, "to the extent necessary, actions should be implemented to prevent the terrestrial digital transition from leading to further or undue concentration of ownership or control of media. Some possible measures are the adoption of regulatory provisions relating to the operation of multiplexers, clear rules of competition, and pricing for multiplexers, and distribution networks, and differentiation between distribution, and content production operations within the same company, among others".

On the other hand, as pointed out by González-Bustamante and Soto Saldías (2016), although the electronic media of press, radio, and television conglomerates adopt a digital form of pluralism and a market vision of their work, due to their structure, they can both put issues on the agenda, expand, and preserve the presence of official and ideologically close sources while having a greater number of publications. In contrast, the periphery and/or community media place other issues on the agenda but with a smaller thematic breadth.

Similarly, in an exploration of key cases, González-Bustamante and Soto Saldías (2016) highlight the positive action of digital media, presenting smaller agendas that contribute to diversity:

> "(...) the digital media belonging to the large press conglomerates have very similar political agendas, and they address the issues in a critical, and somewhat aggressive way; on the other hand, alternative digital written media, despite concentrating on a few topics—although they address more of them in absolute terms—and presenting less diversity, verify agendas different from those carried out by more traditional media. This suggests that these new media incorporate different themes that would bring diversity to the media system". (González-Bustamante and Soto Saldías 2016)

However, Mellado and Scherman (2020) found less diversity in the media, including those outside the commercial circuit, pointing to the speed or immediacy of traditional media and the lack of resources in peripheral media:

> "The study results also show that online media are not clearly linked to a greater diversity of sources or perspectives. (...) The low level of diversity found in online media can be explained by several elements that characterize how digital media work is done in Chile. In general, the journalistic teams are small, have few resources, live under immense pressure to publish their content as quickly as possible, and make many decisions to maximize readership (number of clicks), all of which might prevent the performance of journalistic work that features diverse representations of the social world daily. This is even more evident in media outlets that form part of non-traditional media and have scarce resources. In the case of Chile, this group includes media that are exclusively digital (as El Mostrador and El Dínamo) and have websites with fewer interactive elements and capabilities such as videos, animations, and live streaming". (Mellado and Scherman 2020)

In this search for pluralism, the scientific literature has focused on digital media due to their ability to address issues other than the media agenda (with another perspective). However, their professional and financial shortcomings do not allow them to tackle a broad agenda: "They address fewer topics, but in a more concentrated way, that is, they focus on some specific topics. On the other hand, the agendas of the big media, although similar, treat the issues more evenly and do not concentrate only on a handful of topics" (González-Bustamante and Soto Saldías 2016). In other words, conglomerates' digital media show homogeneity of content. Conversely, with their thematic limitation, independent electronic media can accompany communities with their demands, exercising—in practice—the right to communication Hasbún-Mancilla et al. (2017).

Faced with the above, quantitative and qualitative lines of evidence is necessary to elucidate a series of issues related to the emergence of commercial, peripheral, or community digital media to guarantee greater pluralism in Chile.[1]

## 4. Methodology and Results

This paper uses a mixed interdisciplinary methodological strategy (researchers in communication sciences for the qualitative approach and legal sciences for the quantitative one). To register the digital media ecosystems and their interaction mechanisms with the traditional information system in shaping Chilean public opinion, semi-structured interviews with key stakeholders were conducted. We selected Twitter to confirm, reinforce, and complement the qualitative results. The selection of this specific social network was based on the possibility to go back and scrape data from various terms and a wide range of users. Furthermore, this allows for the determination of some critical user information such as validation, location, and other information that can validate the reliability of the data (Agarwal et al. 2011; Zhang and Ghorbani 2020). Furthermore, compared with other social media, Twitter has been used in a range of topics for sentiment analysis with proven success (Agarwal et al. 2011; Hutto and Gilbert 2014; Nandwani and Verma 2021; Zhang and Ghorbani 2020).

### 4.1. Qualitative Methodology

To describe Chile's traditional and digital ecosystems in terms of its external and structural pluralism, a data classification instrument was applied to experts by independent judges (N = 30). "Experts" were journalists with a track record of editorial responsibility in media (editors) and academics (journalism) who voluntarily attended the activity and carried out the ad hoc classification of the political orientations of the media based on the partisan categories proposed by Otero (2020) (see Table 1). This author developed a methodology called "AdFontes" that implements a political bias test to rank their left-to-right political positions between 20 policy positions. This methodology allows each media company to create an ideological balance including at least three positions: centrist, left-leaning, and right-leaning analysis. The political positions are based on "bipartisan" positions, understanding centrist position as neutral, right as conservative, and left as mixed liberal, positions from which the rest of the axis is developed The final location of a medium is determined by the average of the scores awarded by the judges (Vernier et al. 2018).

**Table 1.** Media Political orientation.

| |
| --- |
| Most Extreme Left |
| Hyper-Partisan Left |
| Skews Left |
| Neutral or Center Skews Right |
| Hyper-Partisan Right |
| Most Extreme Right |

Source: Otero (2020).

The source typology of Mellado and Scherman (2020) is based on three main aspects of media diversity, in order to determine to what extent source diversity, viewpoint diversity,

number of sources in the news, and dominant source differ in their prevalence in media coverage in order to build a better understanding of pluralism.

Currently, no dataset can have data on media posts, which generates the need to create one through a web-scraping system. These systems have proven reliable for obtaining unstructured data (Contreras 2022; Diouf et al. 2019). For unstructured data collection, an Amazon Web Services (AWS) called Elastic Compute Cloud (EC2) virtual machine was created to host the data crawler system (Chaulagain et al. 2017; Khder 2021). This Python-based system consists of spiders that were used to run through the accounts of 450 Chilean media that pour content into social networks such as Facebook and Twitter (Prastyo et al. 2020; Wongkar and Angdresey 2019). These data were downloaded to be processed offline in the AWS virtual machine, reducing the volume of data processed in the cloud. To avoid being blocked (banned), an IP pool was used so that we could act as users of the website. Patel (2020). Once the shape data were obtained, a machine learning method was applied through which the data obtained from the crawler were analyzed and classified. Some NLP (natural language processing) techniques were also used that allowed us to perform an analysis of feelings, contexts, and others. The search period ran from 1 January 2021 to 31 March 2022. The resulting dataset included approximately 2 million pieces of data and comprised heterogeneous media covering national, regional, and local trends with minor and significant owners.

This corpus was subjected to an automated analysis of entities allowing the identification of the actors mentioned above. After that, it was possible to categorize these actors based on the sources typology proposed by Mellado and Scherman (2020) (see Table 2).

**Table 2.** Source Topology.

| |
| --- |
| State or Political Party |
| Business or Company |
| Police and Security |
| Legal and Court |
| Military or Defense |
| Health |
| Educational |
| Civil Society |
| Religion/Church Ĉitizen |
| Media |
| Sports |
| Performer, Artist, or Celebrity |
| Anonymous |
| Other Sources |

Source: Mellado and Scherman (2020) https://www.journalisticperformance.org/ (accessed on 21 December 2022).

Semi-structured interviews were conducted to propose elements of discussion for a constitutional debate that would facilitate the definition of a plural media ecosystem from a legal point of view. These interviews were conducted with constitutional experts, communicators, and media directors to discuss the normative reality of the digital media ecosystem.

### 4.1.1. Data Collection Techniques

In this study, the proposed qualitative component involved the design, application, and analysis of semi-structured individual interviews. The semi-structured interview corresponds to a primary data collection technique that seeks to obtain information from the perspective of the actors whose life experiences and knowledge are relevant to observe and understand the phenomenon under study. The use of this technique provides contextualized and comprehensive information through the answers and stories of the interviewees themselves. The characteristics of this technique enable each of the answers to serve as a guideline that is explored in an unstructured way (not prepared in advance but systematic). Furthermore, it allows for an investigation of the aspects derived from the

answers provided by interviewees in an environment of trust. By exploring the strongest ideas of interviewees, their behavioral meaning may be deduced.

### 4.1.2. Sample

The selected sample included the following participants:

- Two constitutional experts;
- Two experts in the area of communication;
- Two traditional commercial media directors;
- Two traditional independent digital media directors.

The interview schedule included the following items:

- Pluralism: understanding and exercise;
- Communication law: understanding and clarity/confusion concept;
- Ideology of the environment and other media: political orientation;
- Agenda construction: information selection method;
- Method of collecting information and sources on the constituent process.

### 4.2. Quantitative Evidence

The data flow of the process through which the content of the tweets concerning the country's constitutional discussion was analyzed is shown in Figure 1:

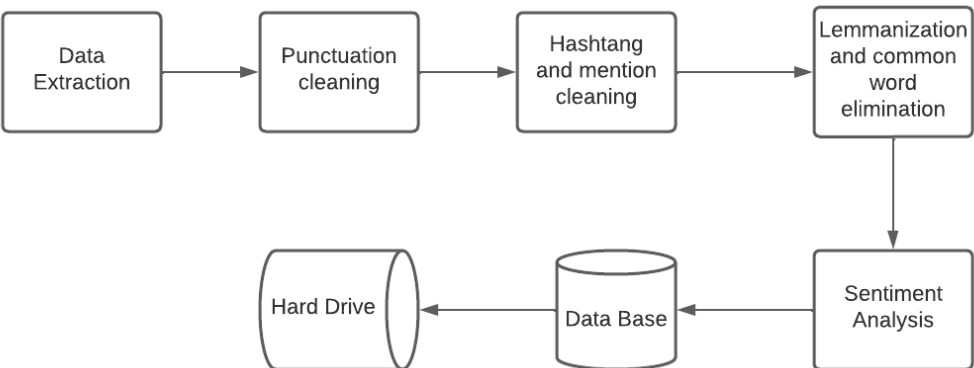

**Figure 1.** Number of tweets per month shows the total in thousands of posts.

As noted, the first step was to use a web-scraping tool. The twarc tool was used along with the academic Twitter API. This combination allowed us to perform up to 10 million tweet extractions per month without being blocked. This tool has been previously used in social science studies similar to this one (Maemura et al. 2016; Ruest and Milligan 2016). The advantage of the twarc tool is allowing researchers to quickly and accurately obtain all the information about the tweet, its author, whether or not it was reposted, and if it could be sensitive. For each hashtag, a CSV file was created that contained all the information of the tweets for later text preprocessing. First, punctuation marks, hashtags, mentions, and common words such as articles and pronouns were eliminated for better analysis of repetition and feeling.

The sentiment analysis was performed using PySentimiento Pérez et al. (2021). One of the greatest difficulties in sentiment analysis is that few tools work natively in Spanish. The most used sentiment analysis tools are Vader and Textblob; however, these require the original text be translated, and therefore, a degree of precision is lost when performing sentiment analysis. Finally, all the files were amassed to create a database for reference and verification.

### 4.3. Data Description

A universe of 122 hashtags and a list of digital and traditional media were defined. Subsequently, we searched for posts related to those hashtags from 1 January 2021 to

31 March 2022. This time period was selected because it spanned from the start of the constitutional discussion until three months after the presidential election. The result was a total of 8,549,245 tweets for the defined period. Figure 2 shows the distribution per month of all the posts related to hashtags. The results show that the period from July 2021 to February 2022 had the largest number of posts (82%), with December 2021 having the largest number of posts (22.9% of the total posts).

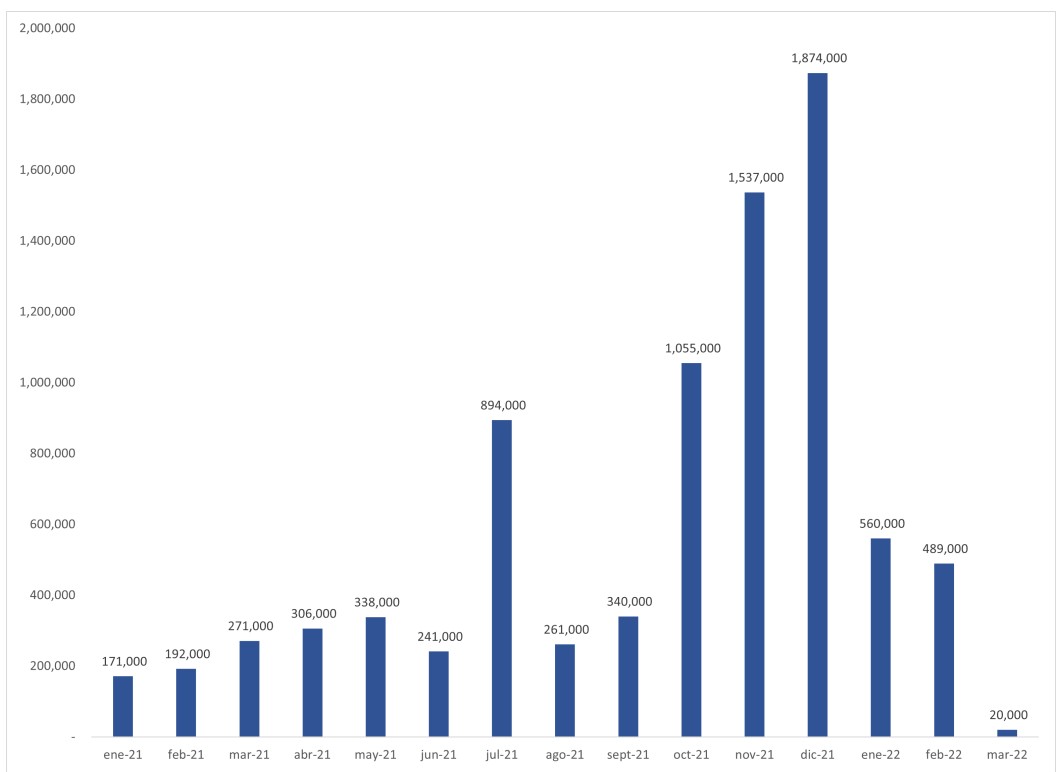

**Figure 2.** Number of tweets per month.

Figure 3 shows the distribution of the posts generated only by the selected media, totaling unique tweets for the defined period. The results show that the distribution is very similar to that of the general sample. However, there was a difference during the first quarter of 2022, when a greater number of posts were generated compared with the general sample.

Figure 4 shows the distribution of hashtags for the overall sample. It can be seen that "Constitutional Convention" only occupied the fourth place, and "Constitutional Convention" (with and without accent) in fifth and sixth place, respectively, and "New Constitution" in the tenth place. On the other hand, the presidential candidates "Boric" and "Kast" and the hashtag "Chile" covered a significant proportion of posts (41% of the total).

Figure 5 shows the distribution of hashtags for the formal and informal media samples. Unlike the general sample, in the digital media sample, the hashtags related to the "Constitutional Convention" were ranked in the first three places. It should be noted that "New Constitution" did not appear as it did in the general total, but it appeared in the concept of "social outbreak", which was not in the ten most recurrent hashtags.

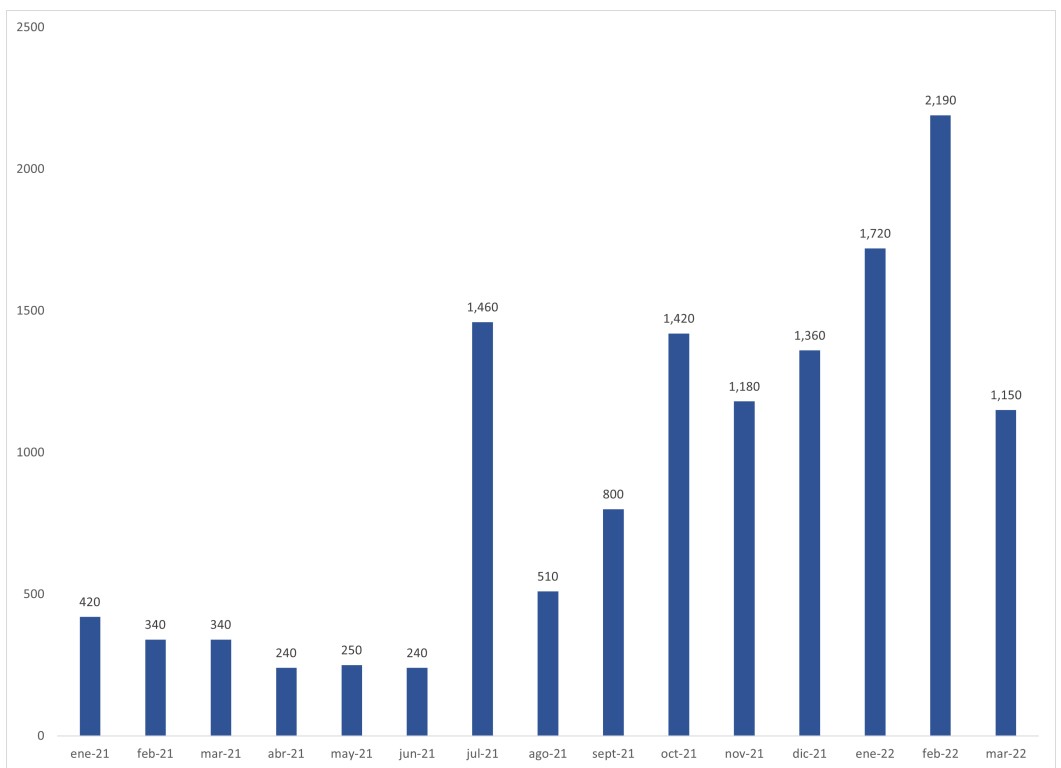

**Figure 3.** Number of tweets per month for media sample.

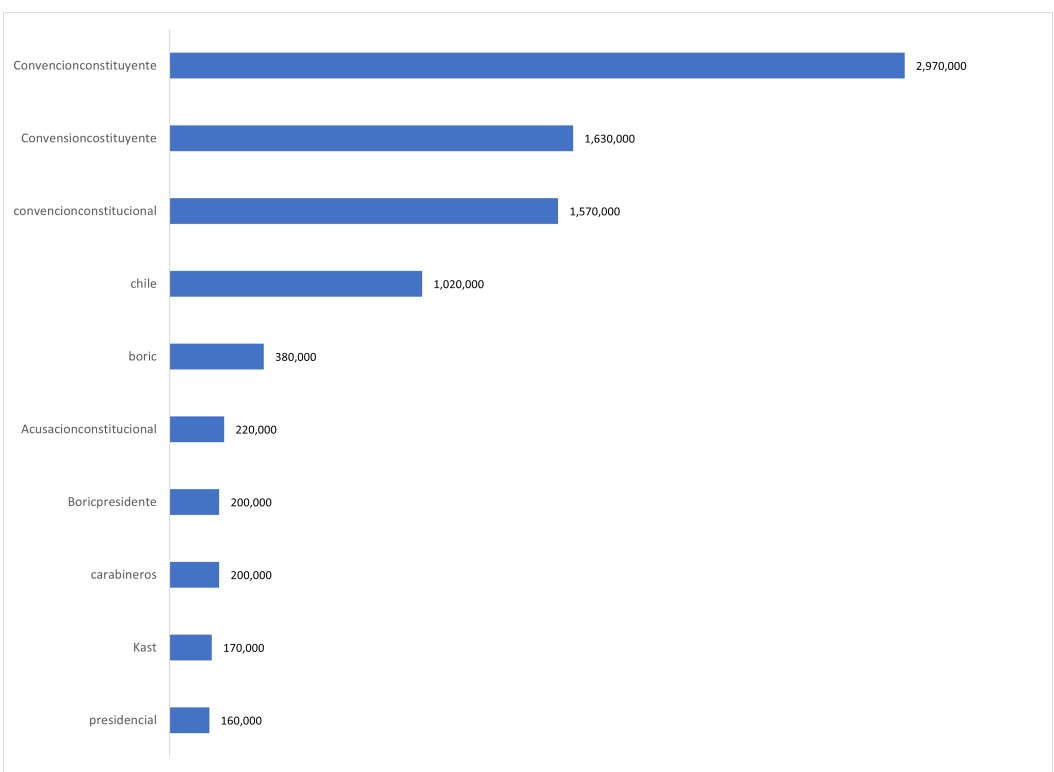

**Figure 4.** Most popular hashtags of the total sample January 2021–March 2022.

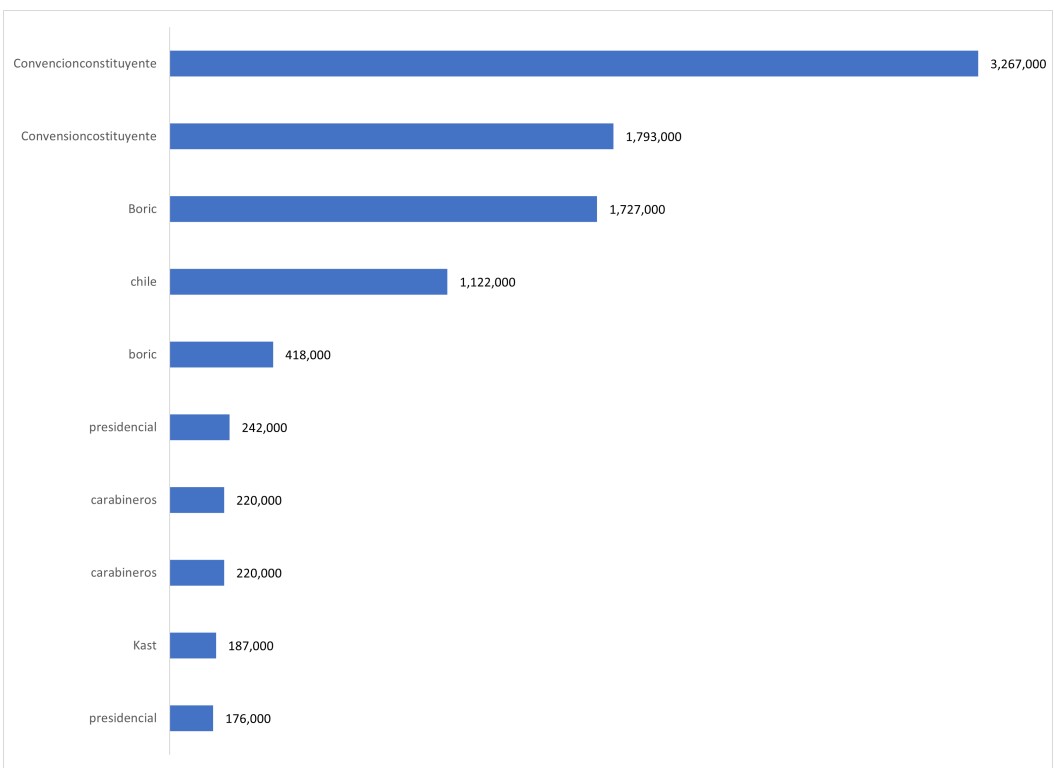

**Figure 5.** Most popular hashtags for media January 2021–March 2022.

Next, a word cloud (Word Cloud) database was developed to present a series of words or labels graphically with different colors and sizes depending on the word's relevance. A maximum of 25 words were taken for each sample (total and medium). Figure 6 shows the general sample, where it can be seen that words such as "government", "people", "vote", and "better" are repeated in greater quantity, agreeing with the most recurrent hashtags. These concepts were closely related to the electoral process taking place in Chile. It should be noted that the word "right" appeared repetitively.

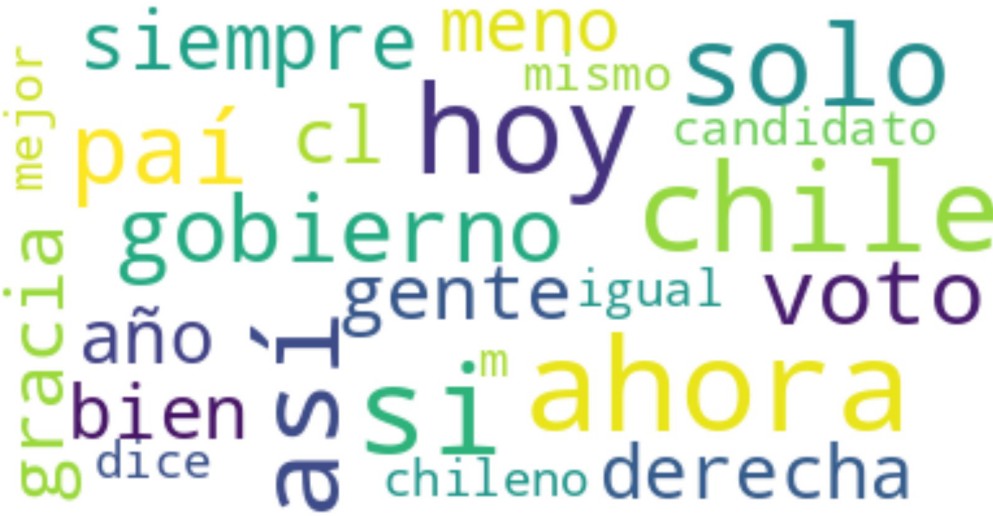

**Figure 6.** Top words for the total sample in the word cloud.

As for the number of repetitions per word, it was observed that both "Chile", and "Sí" ("Yes") had an important weight with respect to the total number of words. "Right", "Government", and "People", which are concepts associated with politics, raised considerable interest (see Figure 7).

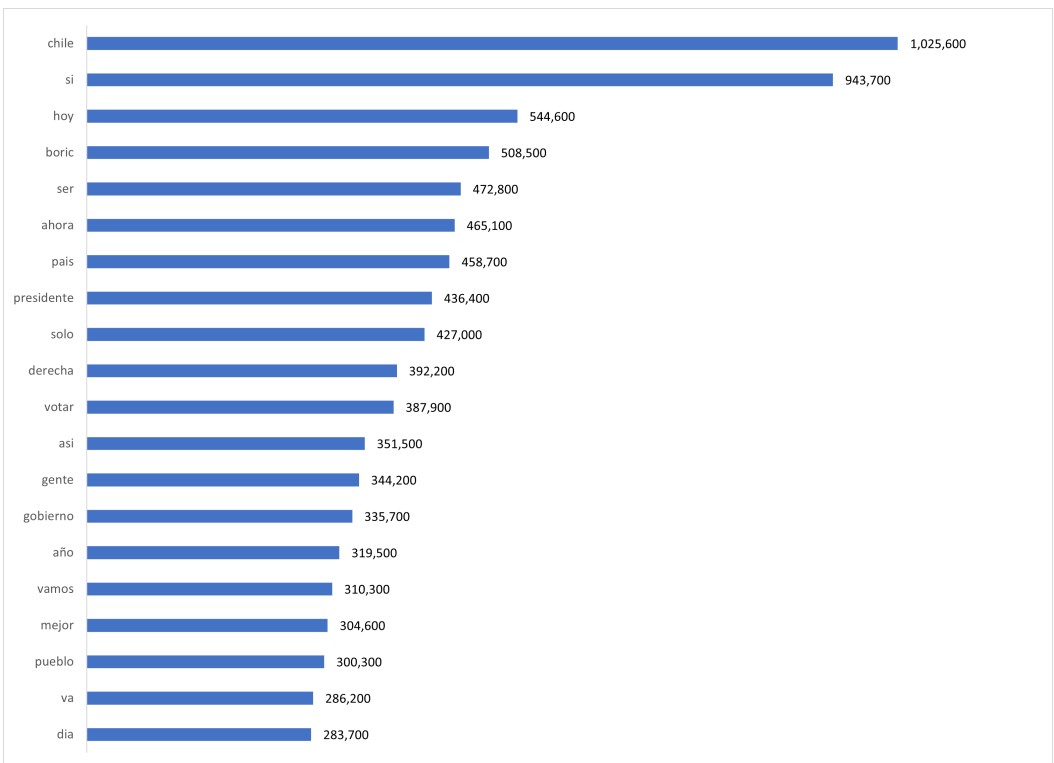

**Figure 7.** Top 20 popular words during the period January 2021–March 2022.

In terms of the media, it can be seen that the words showing the greatest number of appearances were "abstention", "conventional", "constituent", "right", and "carabineros". The first three involve the process of drafting the new Constitution, and the latter is related to the social outbreak that occurred in Chile before the constituent process (see Figure 8).

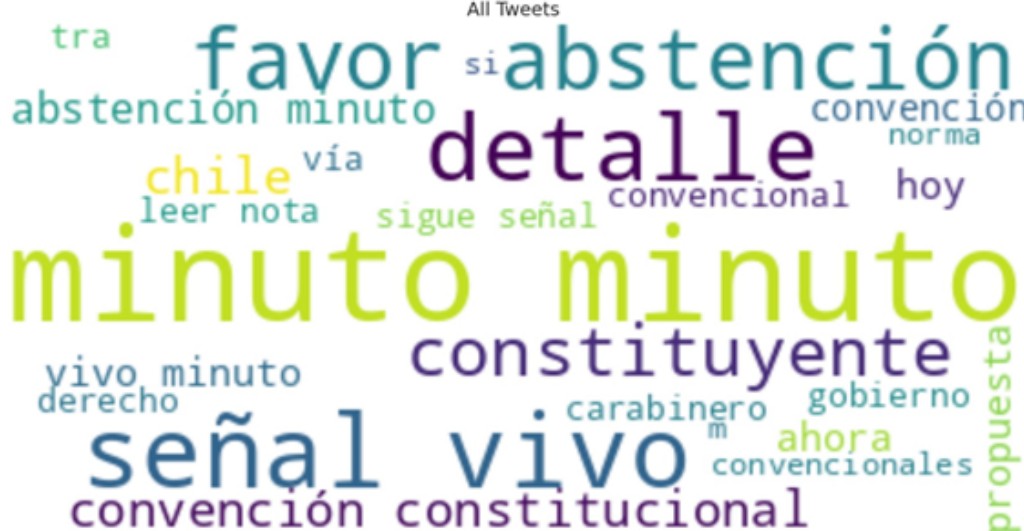

**Figure 8.** Top words for the total sample in the cloud.

Regarding the number of repetitions per word, it can be seen that the most frequent words in the general sample did not have the same distribution in the media sample. "Minute", "Alive", "Convention", "Constitutional", and "Constituent" were among those with the greatest representation. "Chile" was ranked sixth compared with ranking first place in the overall sample (see Figure 9).

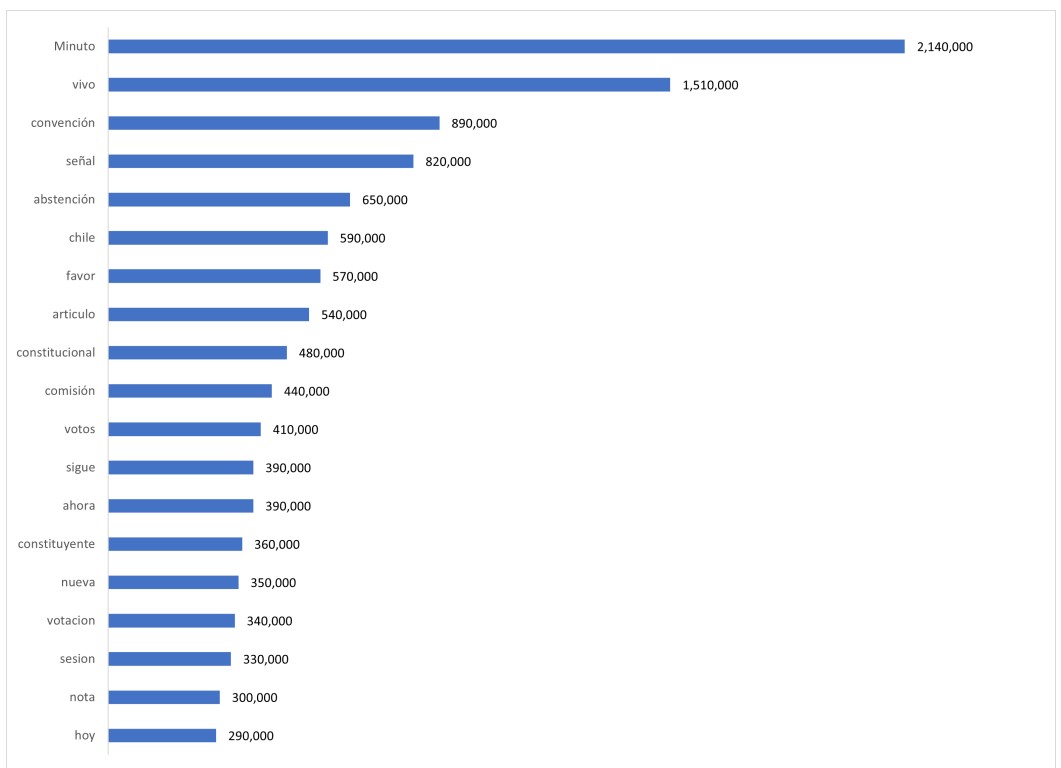

**Figure 9.** Top 20 popular words during the period January 2021–March 2022.

### 4.4. Sentiment Analysis

Sentiment analysis, also known as opinion mining, is a task of automatic mass classification of documents, which focuses on cataloging documents according to their positive or negative language connotation. As mentioned in the previous section, it was difficult to conduct this analysis using a tool with a Spanish corpus in order to predict the feeling of the text more accurately. Media-related tweets were sentimentally analyzed and classified as positive, neutral, and negative.

Figure 10 shows that as the date of the elections approached, negative sentiment grew from 10% in July 2021 to 47% in February 2022, representing a 370% growth. As for neutral sentiment, it decreased from 63% to 49%, representing a decrease of 22%. Finally, in general, in this sample, a high positive feeling was not observed; however, it decreased by more than half, reducing from 9% to 4%. Similarly, Figure 11 illustrates the percentage change in the sentiment level. Positive sentiment had the greatest fluctuation from month to month, with the largest fluctuations close to the election and close to the first drafts. In the case of negative sentiment, during the elections, it did not change much; however, negative growth had its highest level in March 2022, similar to the rejection it had at the beginning of the constituent process. In the case of neutral sentiment, it had low variability.

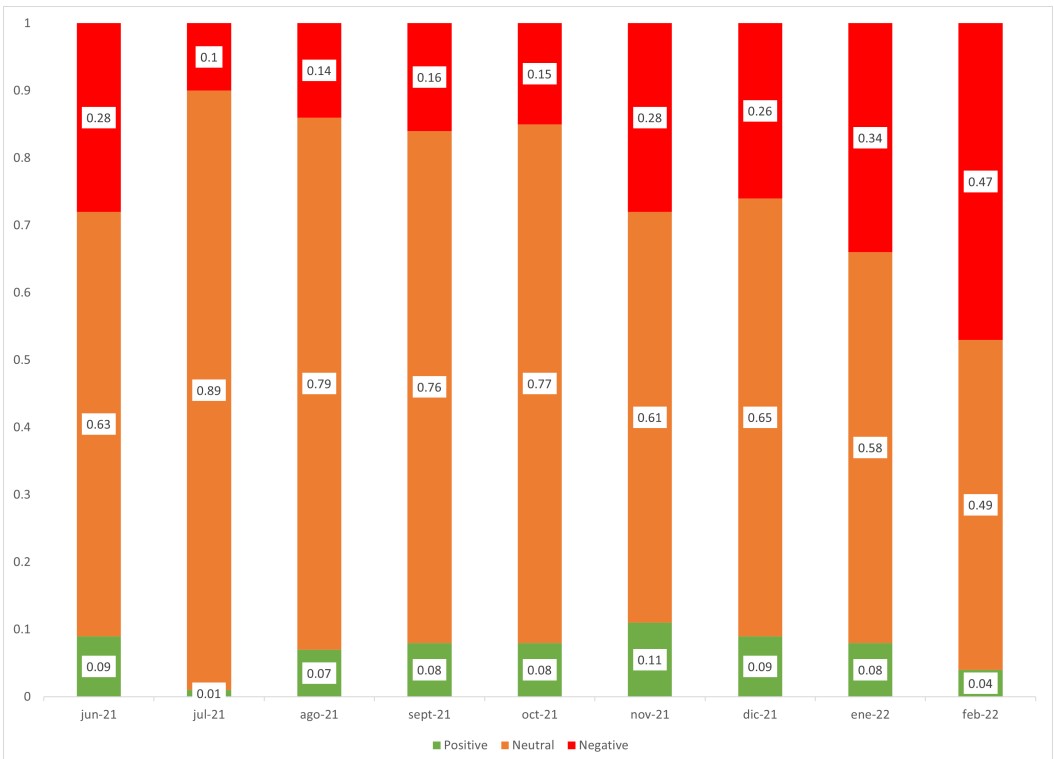

**Figure 10.** Sentiment distribution per month.

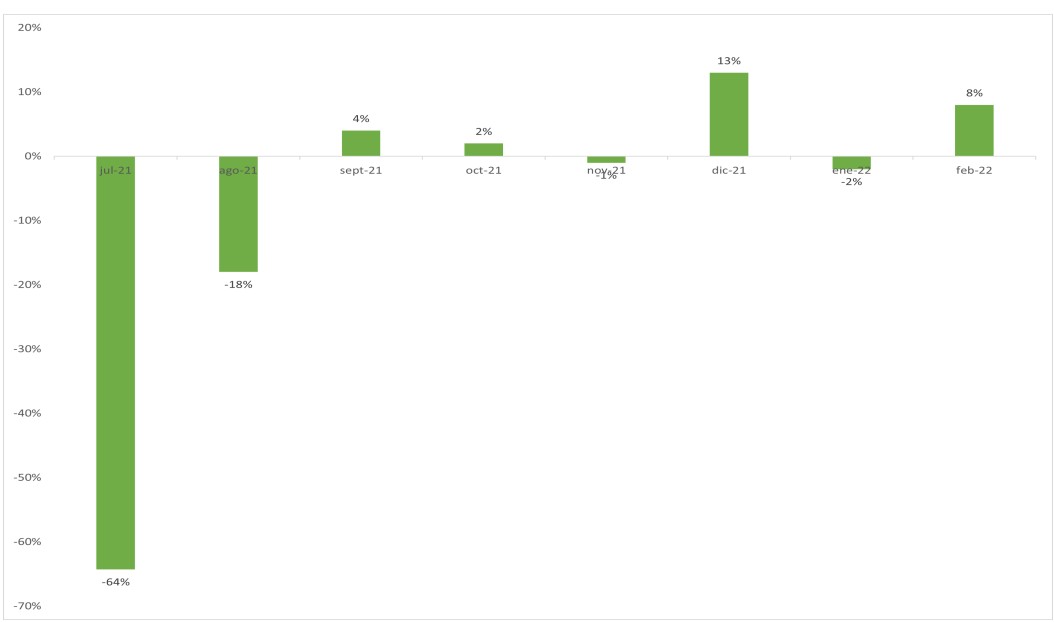

(**a**) Negative

**Figure 11.** *Cont.*

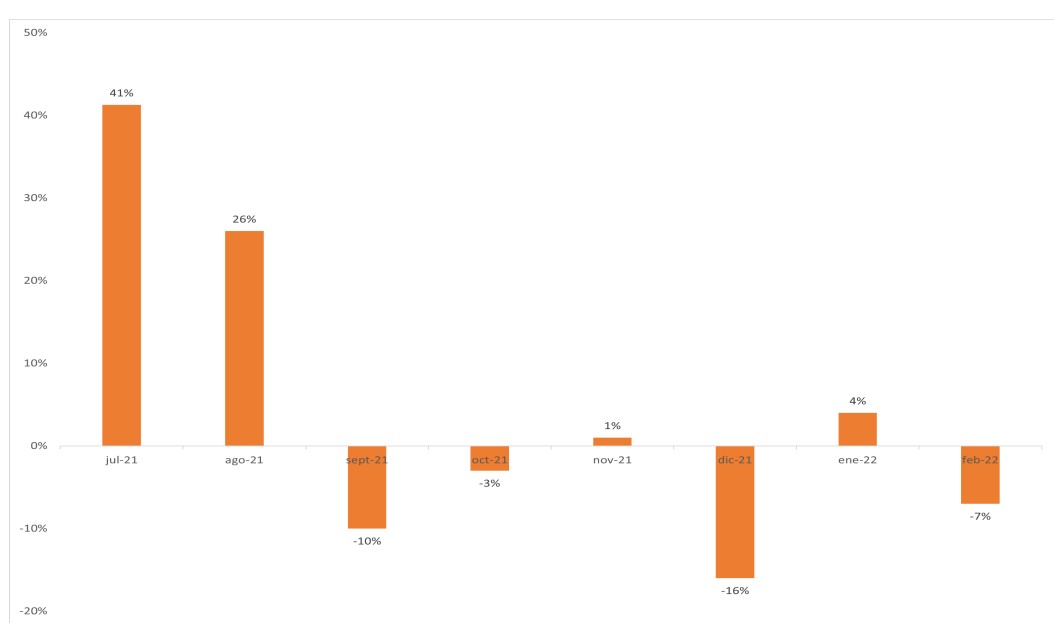

(**b**) Neutral

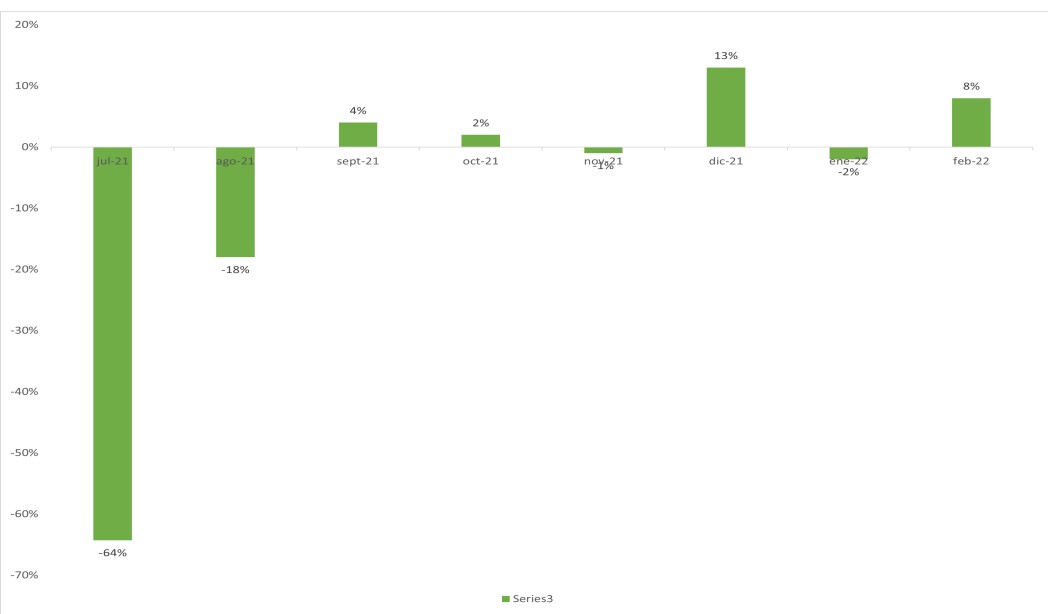

(**c**) Positive

**Figure 11.** Percentage change in negative, neutral, and positive sentiment over time.

Media Sentiment

Each of the media outlets was analyzed based on the type of sentiment their tweets generated. Overall, the level of negativity was 25%, neutrality 65%, and positive 10% (see Figure 12).

Traditional television channels (C13, TVN, MEGA, CHV, RedTV, and CNNChile) revealed levels of negativity, neutrality, and positivity of tweets of 16%, 64.4%, and 19.6%, respectively. RedTv showed the highest level of negativity in the group (24% vs. 16%). At the other extreme, CHV had the lowest level of negative sentiment in its tweets (3.5%). Canal 13 was the medium that had the highest level of neutrality of traditional television channels (86%), while RedTV only had 39.8%. In the case of positive tweets, RedTV and Canal 13 had the highest number and similar levels (36.5% each).

Traditional radios (ADN Radio, Duna, Biobío, Cooperativa, and Agricultura) had levels of negativity, neutrality, and positivity of 34.2%, 56.5%, and 9.3%, respectively. The

level of negative tweets was higher than the median average (34.2% vs. 25%). The medium with the highest level of negative tweets was ADN Radio, with 65.97%, followed by Radio Biobío (39%). Agricultura radio had the highest level of neutral tweets (85.8%), and ADN Radio had the highest proportion of positive tweets (25.25%).

For written, and/or electronic media (72.3% vs. 56.5%), these were more neutral than radio and television (72.3% vs. 64.4%). It should be noted that their level of negativity was 21%, and positivity was at 6%. El Mostrador showed the highest level of negative tweets (27.15%), while La Cuarta only had 1% of positive tweets, yet a level of neutrality of 82.7%.

The rest of the non-traditional media showed higher levels of negative (29.68% vs. 25%) and positive (15% vs. 10%) tweets than the rest, along with lower levels of neutrality (56% vs. 64%). In the case of Mi Radio, LS had a level of negative tweets close to 96%, which, together with Crónica Chile (79%), had the highest number of negative tweets. UATV (97%), Tres Quintos (96%), and El Morrocotudo (84%) showed the highest levels of positivity. Finally, Terra (78.4%), El Líbero (50%), and Crónica Chile (20%) had the highest levels of positive feelings.

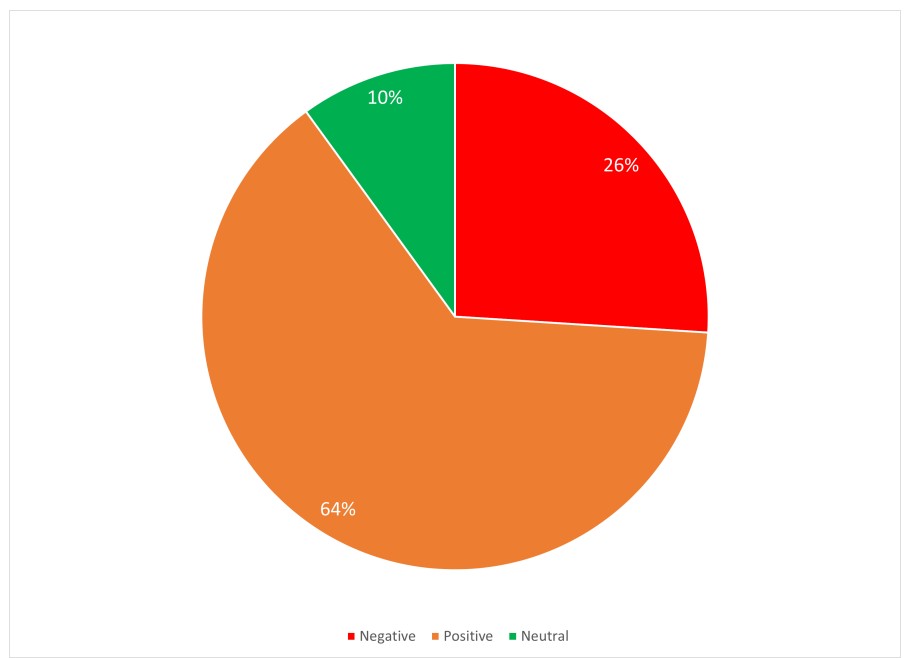

**Figure 12.** Sentiment ratio, media sample.

*4.5. Qualitative Methodology for Interviews*

The project design involved the implementation of a qualitative component of a descriptive–explanatory type, whose central characteristic is to seek in-depth knowledge of the social and experiential reality of a relevant sample of actors linked to the investigated theme, using the online qualitative tools. resulting from the first part of this methodological application. The analysis, therefore, was carried out through an inductive process, starting from the qualitative textual data, rather than a deductive process. Thus, the analysis revealed experiences and meanings through access to discourses that carry a historical, political, and cultural substrate.

4.5.1. Data Collection Techniques

The proposed qualitative component involved the design, application, and analysis of semi-structured interviews of an individual nature. The protocol to carry out the interviews was online (Zoom), conducted individually, lasting one hour each, and carried out between the months of July and September of 2021. The application of the interviews consisted of a set of general questions as follows:

- Does the emergence of commercial, peripheral, or community digital media guarantee greater pluralism in Chile?
- What are the regulatory proposals and their limitations to generate media decentralization in relation to digital media?
- Do you consider that the process of establishing the Constitutional Convention has been transparent? Have people had plural access to the issues addressed in the Convention and the news derived from its work?
- How could these regulatory innovations be linked to a new Constitution and, specifically, to the constitutionalization of the right to communication?

These general questions were further developed in terms of the following specific items:

- Pluralism: understanding and exercise;
- Communication law: understanding and clarity/confusion concept;
- Ideology of the environment and other media: political orientation;
- Agenda construction: information selection method;
- Method of collecting information and sources on the constituent process.

The semi-structured interview is a primary information collection technique through which information is obtained from the perspective of actors whose life experiences and knowledge are important to observe and understand a phenomenon. The use of this technique provides contextualized and comprehensive information through the answers and stories of the interviewees themselves.

The characteristics of this technique enable each of the answers to serve as a guideline. that is explored in an unstructured way (not prepared in advance but systematic) and investigate the aspects derived from the answers provided by the interviewee in an environment of trust. By exploring the strongest ideas of interviewees, their behavioral meaning may be deduced.

4.5.2. Sample

The selected sample included the following actors:

- Two constitutional experts;
- Two experts in the area of communication;
- Two directors of traditional commercial media;
- Two traditional independent digital media directors.

The interview schedule included the following items:

- Pluralism: understanding, and exercise;
- Communication law: understanding and clarity/confusion concept;
- Ideology of the environment and other media: political orientation;
- Agenda construction: method of selecting information;
- Method of collecting information and sources on the constituent process.

4.5.3. Qualitative Evidence for Interviews

The constitutional experts interviewed agreed on the contradictory imposition that the claim indicating that the state is the arbiter and owner of the truth seems complex. They advocated for pluralistic free competition, especially in digital media. "I do not know if, I do not remember well, but I don't think there are many constitutions in the world that collect or demand from the media system (or linked to the right to information) the issue of pluralism. I believe that this is fully delivered to the law". Regardless of its format, deconcentration concerning digital media is achieved through competition rules or laws. In terms of competition, some criteria make it possible to identify if a particular market is concentrated; therefore, if that occurs, they force deconcentration. At the same time, one of the interviewees referred to the need to regulate these matters more strictly in the digital field, as opposed to regulations of other markets. Another alternative could be the existence of public media.

On the other hand, in the field of expert actors in the communication of national universities, the interventions were categorical enough to call for the need to stop not only information hegemony but also socioeconomic hegemony. The effect is that it turns the traditional media into allies of the business powers and, in turn, of the political powers behind these business powers, going far beyond the informative, an issue that was installed and imposed in the dictatorship. The ability to reverse it should be a fundamental task within the constituent process: "I believe that finally the issue of ownership and diversity of property in the media also has to do with the very conception of the state (...). If we recognize a plurinational state, the decentralization, and deconcentration of power, we must also consider how that falls on the digital media ecosystem".

The directors of traditional commercial media expressed the need to deconcentrate because there must be more perspectives, including the digital version of these: "I understand that talking about media deconcentration sounds good, and it is a positive question when you say why so many radios are in the hands of a single controller, or why three groups have been articulated, or four chains of radio groups, and there is no greater diversity. As far as I know, it is a difficult business to survive if you have only one medium of communication, a single radio".

Finally, traditional independent digital media directors indicated that, for media digitalization, media law is fundamental: "If you are a banker, or a brewer, or a miner, you cannot have a media outlet" since the conflicts of interest are apparent. "You can be a billionaire: Juan Sutil always wants to buy a medium... The idea would be, if you wish, to buy a medium, but you must get rid of your non-media companies". "The intersection between commercial interests, and, in addition, press owners is the most harmful thing there is for democracy". "The idea is to raise a media ownership law; a second point that has been struggling for years, and no one has given a solution beyond good words, is the deconcentration of the state's notice, the state, not the government, the state is the largest advertiser in Chile. In general terms, we need a law with media ownership, transparency in distributing the state's notice, and a mini constellation of newspapers and state digital media financed by the state".

## 5. Conclusions

The methodological results of a quantitative nature allowed us to notice an increase in information pluralism through digital media that translates into greater political pluralism in shaping public opinion. For example, we noticed a variety of dominant concepts in the media that did not have the same representation in social networks (for example, the concept of "social outbreak" that appeared under-represented (Figure 4)). The country's political contingency was reflected in the search criteria on social networks (Figure 6). The number of repetitions confirmed this by the words used with greater frequency in the general sample of social networks (Figure 8), which did not have the same distribution in the media sample.

On the other hand, social networks allowed us to investigate the variations in users' political feelings (Figure 10), in which different behavioral patterns concerning the media were observed. In particular, political feelings (positive or negative) varied, especially in social networks, compared with the editorial line of the different media. For example, negativity was higher in the editorial line of some media compared with others. (Figure 11). Variation also occurred depending on whether the medium was radio or television (Figure 12). In short, this implies that the media editorial lines have a relevant impact on political feelings.

Interviewing constitutional experts led us to contradictory conclusions. On the one hand, in the competition among information companies and their regulatory frameworks, interviewees recognized an essential component of media deconcentration, which includes the irruption of digital media. In this context, it should be acknowledged that the massive irruption of digital media is favored for market reasons: shallow entry barriers and very low or free transaction costs for the public. However, this market's regulatory framework based on information pluralism still needs to be revised.

Interviews with the expert actors in communication revealed that the constituent process is important for generating the terms and conditions to break media concentration by limiting media ownership. On the other hand, interviews with the directors of independent digital media aimed to give voice to those who are precisely expanding the information spectrum beyond traditional margins. The interviewees proposed that information companies have a single line of business to avoid being part of the holdings of prominent entrepreneurs who try to influence public opinion. This would be part of a media ownership law. On the other hand, they stressed the importance of distributing the state's advertisement with an important proportion to digital media since this economic income is the most important for this media type.

Based on the above, a series of normative innovations are proposed below, considering that pluralism, concentration, and the emergence of digital media require a legal framework that offers civil society the opportunity to exercise its rights of expression and information effectively:

Indeed, Chile has issued a series of regulations seeking to deepen information pluralism on new technologies. Thus, Law No. 18,168 on the General Telecommunications Law establishes a general framework concerning different types of media, which includes new technologies under the broad concept of "telecommunications" while establishing a guarantee of access of all inhabitants to the telecommunication means available in the country. Law No. 20,808 on free choice of telecommunication services, enacted in January 2015, solves a recurring problem in buildings with community departments, which is the obligation to hire a certain telecommunication service provider, guaranteeing free access for owners to choose any plan or offer available; thus, agreements that prohibit the entry of other telecommunication service providers to the building are unenforceable.

However, our position is that there are still many regulatory difficulties in these new technologies that undermine due pluralism. The original draft of the Constitution of 1980 envisaged a National Council of Social Media. However, the formula finally imposed by the constituent assembly in Article 19 No. 12 was the existence of an autonomous body only on television, that is, the National Television Council, omitting radio, which was submitted to the Undersecretariat of Telecommunications (Subtel) of the Ministry of Transport and Telecommunications. Of course, this affects the independence and impartiality of a public service endowed with autonomy (regardless of the government in power) to grant a concession and exercise the regulatory, supervisory, and sanctioning powers in radio, digital media, and new technologies that are also subject to the technical superintendence of the Subtel Balbontín and Maldonado (2019).

Concerning Internet regulation, our legislation does not contain a general regulatory framework on the operation and provisions of the Internet, which is why the provisions of Law No. 18,168 on telecommunications apply generically. There are only specific laws in our legislation that address individual problems concerning the Internet, such as net neutrality or the guarantee of a minimum speed to browse. As a result, there is a regulatory vacuum regarding how social media shapes public opinion. This is inseparable from the fact that the spectrum for which concession is more likely. became exposed to purely being considered a market good, so we can hardly think that, in the current state of affairs, information pluralism through new technologies is close if they exercise economic control that may affect, in turn, the political pluralism in the country. The latter has not been properly investigated.

Although the emergence of digital media is favored for market reasons with lower transaction costs than traditional media, attention must be paid to the existence of invisible or emerging entry barriers through the new technological trends arising in the world, as well as to the entry of new competitors offering technological services. As Mendel et al. (2017) pointed out in the 2013 Joint Declaration on the protection of freedom of expression and diversity in the terrestrial digital transition, "To the extent necessary, actions must be implemented to prevent the terrestrial digital transition from causing a greater or undue concentration of ownership or control of the media. Some possible



measures are the adoption of regulatory provisions regarding the operation of multiplexers, clear rules of competition, pricing for multiplexers, and distribution networks, and differentiation between distribution and content production operations within the same company, among others" (Mendel et al. (2017), p. 25). In short, the problems arising from the interaction of users with each other should be investigated not only for shaping the political public opinion but also for the interaction of users versus the participation of Internet service providers.

The Internet is a particular niche that could facilitate the massification of hate speech against specific people or groups of people with particular characteristics that generate self-censorship. Part of the function of information pluralism is to allow a diversity of opinions and participation of political actors in the same communication channel; however, manifestations of hatred tend to destroy both opinions and even the people who issue them, limiting the pluralism of political positions. Although it is impossible to exercise some control over the opinions issued, nor can there be censorship to prohibit those opinions, regulatory programs must be established between states and Internet service providers to prevent opinions that affect the proper pluralism of information.

In summary, our legislation is still in the initial phase regarding the regulation of digital media and the Internet. Consequently, there are still general regulatory frameworks regarding the Internet and new technologies that must be specifically developed to ensure information pluralism and, ultimately, favor political pluralism in democracy.

**Author Contributions:** All Authors contributed equally to the paper. All authors have read and agreed to the published version of the manuscript.

**Funding:** This research was funded by the National Agency for Research and Development (ANID) Chile with Program Estudios sobre Pluralismo en el Sistema Informativo Nacional, grant number 210009.

**Conflicts of Interest:** The authors declare no conflict of interest.

## Note

1 For a better understanding of the context of digital media in Chile regarding its appearance, evolution, and current situation: Cfr. (Balbontín and Maldonado 2019), chapter IX "Digital media and technologies of information", pp. 621–34.

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
