# Peer review of "Using Sentiment Analysis in Understanding the Information and Political Pluralism under the Chilean New Constitution Discussion"

_socsci, doi:10.3390/socsci12030140_

Round 1

Reviewer 1 Report

The proposed article is in line with the expectations of the editors of the special issue. They expected papers whose theoretical, methodological, multidisciplinary, and substantive approaches advance and analyze journalism ethics practices around the world. They are interested in scientific papers which focus on the analysis, study, and description of media ethics in the widest spectrum of professional practices.

Presented paper focused on media and human rights, media ethics, media and law, new technologies in journalism and communications, political behavior in the digital public space.

In keeping with the title “Using Sentiment Analysis in Understanding the Information and Political Pluralism under the Chilean New Constitution Discussion, “the study aims to investigate the forms of concentration such as the communicational flow of digital media to determine the opportunities and information defects of these media due to regulatory deficiencies in this sector. However, for the clarity of the text, it would be useful to specify the research goals more precisely, and especially the unambiguous questions that the authors answer.

In the first part, I propose to put more emphasis on why the authors chose these research methods. It would be particularly important to explain how the measurement of sentiment and keywords can bring us closer to the importance of new media in the fight for pluralism and the fight against concentration and monopolism. To what extent can quantitative methods show the necessity of constitutional changes?

The authors write about the risk of concentration and the lack of media pluralism in Chile, the article lacks empirical data showing the degree of media concentration and the scale of the real threat to media freedom.

In the first part, it would be possible to define the terms of digital media based on literature. In sentence 103, they seem to be synonymous with social media.

In the graphic layer, it would be clearer if the graphs were next to their descriptions. It is worth adding that some graphs, e.g., 4, 5, 7, 9 are illegible. In addition, the charts are in Spanish, which, especially in the case of the Tag Cloud, makes it illegible for a non-Spanish reader.

 In the summary, it can be emphasized once more clearly how the analysis of sentiment and the frequency of occurrence of words brings us closer to understanding the Information and political pluralism under the Chilean new constitution discussion.

Reviewer 2 Report

General commentaries

This article presents an analysis of the public debate on the new constitution in Chile. Contributions: 1) the debate from the perspective of information and political pluralism; 2) the debate in digital media and social media; 3) incorporate the variable "sentiment analysis" on social media. It provides valid documentation and various lines of future research.

Various investigations are cited in the article. The legal context is interesting and timely. But it would be opportune to have the context of digital media in Chile, some data that facilitate the understanding of the subject. A brief exposition of key dates of the appearance, evolution, and current situation of digital media in Chile would also be opportune. Situation of social media in Chile.

Another weakness is the absence of a systematic review of research on political pluralism, “sentiment analysis”, “social networks and political pluralism” and “social networks and public opinion”. It would be convenient for the author/s to include this type of research.

Expand the characteristics methodologies used. Specify the research questions. What were the semi-structured questions?

The article is clear, relevant, the structure can be expanded.

There are bibliographical references cited from recent publications. There are few cited references to scientific research articles that delve into some of the topics of the article. For example: “sentiment”. It can improve.

The methodology needs to be expanded. This point can be improved. Contribute: criteria for the selection of the sample. An analysis period is indicated: what are the selection criteria? The profiles for the semi-structured interviews are provided: what are the criteria for choosing these profiles?

Some images are low resolution. It’s difficult to read.

The absence of scientific research and analysis on issues such as "sentiment analysis", "social networks and political pluralism" and "social networks and public opinion" leads to conclusions that do not deepen.

Specific commentaries

Figure 4. Poor quality image.

Figure 5. Poor quality image.

Figure 6. Improve image resolution.

Figure 7. Improve image resolution.

Figure 8. Improve image resolution.

Figure 9. Improve image resolution.

Figure 11. Improve image resolution.

Figure 12. Improve image resolution.

Table 1. The title is in Spanish. It is written “source”, it should be “Source”.

Line 181: "categories proposed by Otero (2020)". It would be appropriate to include in the table a brief description of the categories.

Line 201: "typology of sources proposed by Mellado and Scherman (2020)". It would be appropriate to include in the table a brief description of the typologies.

Line 220. It would be opportune to know the criteria for the selection of the sample.

Line 252. It would be appropriate to know the criteria for the selection of hashtags and the media.

Line 255. What were the reasons for choosing that period?

Line 293. The sentiment analysis is on Twitter. What are the scientific arguments for choosing this social network?

Line 347. What were the semi-structured questions asked? What were the criteria for the selection of interviewees? Was there any protocol in the interview?

Round 2

Reviewer 2 Report

Thanks for the corrections and new content to the article.